# NOISE REDUCTION IN BERT NER MODELS FOR CLINICAL ENTITY EXTRACTION

## ABSTRACT

Precision is of utmost importance in the realm of clinical entity extraction from clinical notes and reports. Encoder Models fine-tuned for Named Entity Recognition (NER) are an efficient choice for this purpose, as they don't hallucinate. We pre-trained an in-house BERT over clinical data and then fine-tuned it for NER. These models performed well on recall but could not close upon the high precision range, needed for clinical models. To address this challenge, we developed a Noise Removal model that refines the output of NER. The NER model assigns token-level entity tags along with probability scores for each token. Our Noise Removal (NR) model then analyzes these probability sequences and classifies predictions as either weak or strong. A naïve approach might involve filtering predictions based on low probability values; however, this method is unreliable. Owing to the characteristics of the SoftMax function, Transformer based architectures often assign disproportionately high confidence scores even to uncertain or weak predictions, making simple thresholding ineffective. To address this issue, we adopted a supervised modeling strategy in which the NR model leverages advanced features such as the Probability Density Map (PDM). The PDM captures the Semantic-Pull effect observed within Transformer embeddings, an effect that manifests in the probability distributions of NER class predictions across token sequences. This approach enables the model to classify predictions as weak or strong with significantly improved accuracy. With these NR models we were able to reduce False Positives across various clinical NER models by 50% to 90%.

## 1 INTRODUCTION

In the realm of Precision Medicine, data that completely and accurately captures complex concepts such as mutational status of a tumor, stage or performance status, is critical. The extraction of such entities from unstructured clinical notes, including biomarkers, tumor stages, and performance scores, demands both high precision and high recall. To address this challenge, we developed a high-fidelity clinical Named Entity Recognition (NER) model over BERT (Devlin et al. (2019)) architecture, trained over millions of clinical notes. On this foundational model, we fine-tuned for a NER objective aimed at extracting domain-specific clinical entities. Despite the rigorous training, construction of large-scale language models inherently introduces noise at various stages, owing to ambiguity in real world clinical data, inconsistencies in labeling and sampling-related biases. These issues may propagate during model inference, reducing output reliability. Therefore, a critical component of the pipeline needs to focus upon suppressing noisy predictions to uphold high precision.

NER is inherently a sequence tagging problem, where each token in a sentence is assigned a label with an associated confidence score. It is commonly assumed that incorrect or uncertain predictions correspond to lower confidence scores and can be filtered using a threshold-based approach. However, this method proves inadequate due to characteristics of the Transformer architecture as explained by Vaswani et al. (2017). Specifically, the final prediction layer employs a SoftMax function, which often yields high-confidence scores, even for erroneous or Out Of Distribution (OOD) predictions, owing to its inherent limitations (Guo et al. (2017), Pearce et al. (2021), Tu et al. (2024)).

Uncertainty estimation in neural networks, particularly Transformers, remains an active area of research. Numerous techniques have been proposed to calibrate model confidence scores, each with trade-offs in applicability and effectiveness (Guo et al. (2017)). Broadly, uncertainty can be cate-

gorized into two types: *aleatoric* (statistical/data-related) and *epistemic* (model-related). Aleatoric uncertainty arises from intrinsic noise or ambiguity in the data, while epistemic uncertainty results from limited knowledge or insufficient training data. Distinguishing between these two forms is non-trivial and both can be reflected, albeit imperfectly, in SoftMax score distributions. While the SoftMax function itself is effective for estimating relative token probabilities (Lokshin & Kreinovich (2023)). It tends to inflate confidence in OOD contexts or under high uncertainty, which can hinder the identification of erroneous outputs.

In the context of NER, our primary challenge was the identification of false predictions. Rather than directly calibrating probability scores, we adopted a pragmatic approach focused on detecting patterns of uncertainty within the model's output. Since individual token probabilities often lack sufficient discriminative power, we analyzed sequences of tokens and their contextual dependencies. Due to positional encoding and the self-attention mechanism, predictions for adjacent tokens in Transformer architectures are inherently interdependent. This interdependence induces a Semantic-Pull effect, wherein semantically related and frequently co-occurring tokens in the training data exhibit high mutual attention and converge in the embedding space.

To leverage this phenomenon, we designed a comprehensive set of probability density and statistical features inspired by the internal dynamics of Transformer models. Specifically, we constructed a Probability Density Map (PDM) to capture the contextual neighborhood of each predicted NER token. This was further augmented with statistical indicators such as inter-class probability differentials, sequence entropy, NER class probability strength, etc. Collectively, these features encapsulate the broader semantic and statistical context governing each prediction, thereby enhancing the model's ability to identify uncertain or erroneous outputs.

We then employed a Decision Tree model to classify each NER prediction as either "strong" or "weak," based on the derived uncertainty features. The Decision Tree was selected for its interpretability, allowing transparent understanding of how uncertainty patterns affect classification. Recognizing the risk that true entities could be mistakenly flagged as weak due to overlapping uncertainty signals, we further refined the decision boundary to minimize False Positives (FPs) while preserving True Positives (TPs) to the greatest extent possible.

This uncertainty-aware post-processing framework provides an effective mechanism for enhancing the precision of NER outputs in clinical NLP pipelines. By integrating model-internal probability signals with statistical heuristics, we achieve a practical balance between recall and precision without modifying the core language model architecture.

## 1.1 RELATED WORK

The challenge of modeling uncertainty in neural networks has been extensively explored due to its widespread occurrence across various architectures, including Convolutional Neural Networks (CNNs), Long Short-Term Memory (LSTM) networks, and Transformers. Researchers have primarily focused on addressing two types of uncertainty: epistemic and aleatoric. Studies have revealed that standard SoftMax-based neural networks are prone to feature collapse and often extrapolate unpredictably when presented with OOD data points.

Two principal schools of thought have emerged in this domain. The first aims to calibrate the SoftMax output probabilities, while the second focuses on devising numerical methods to estimate uncertainty directly.

A significant body of research has concentrated on calibrating SoftMax probabilities to mitigate miscalibration issues. Guo et al. (2017) introduced several evaluation metrics for calibration, including Reliability Diagrams, Expected Calibration Error (ECE), and Negative Log-Likelihood (NLL). They further surveyed popular calibration techniques such as Histogram Binning, Isotonic Regression, Bayesian Binning into Quantiles (BBQ), and Platt Scaling. Their key contribution was the proposal of *Temperature Scaling*, a simple yet effective post-processing method to calibrate SoftMax outputs. *Temperature Scaling* is stated as: $\hat{q}_i = \max_k \sigma_{\text{SM}} \left( \frac{\mathbf{z}_i}{T} \right)^{(k)}$ where $z_i$ is the logit for input $x_i$, and the SoftMax output or predicted probability for class $k$ is denoted by $\sigma_{SM}(z_i)^k$, temperature scaling involves dividing the logits by a temperature parameter $T$ before applying the SoftMax function to adjust the confidence of predictions and generate a calibrated probability $\hat{q}_i$.

Beyond calibration, several works have attempted to enhance the handling of OOD samples using modifications to the SoftMax function. Tu et al. (2024) proposed a method called *Softmax Correlation (SoftmaxCorr)*, which leverages the cosine similarity between predicted probability vectors to construct an inter-class correlation matrix during inference. This matrix is compared with a reference matrix computed on in-distribution data. A higher deviation from the reference implies a greater level of uncertainty. Higher SoftmaxCorr score indicates model confidently assigned probability to the predicted class.

In another advancement, Możejko et al. (2018) introduced Inhibited SoftMax, an extension of the standard SoftMax that includes an additional constant input representing the network's uncertainty. This component is incorporated into the loss function to simultaneously minimize classification error and maximize certainty, thereby explicitly encouraging the model to learn representations reflective of its own confidence.

The second key area of research involves direct estimation of uncertainty. Gal & Ghahramani (2016) made a seminal contribution by demonstrating that Monte Carlo Dropout (MC-Dropout) can serve as a Bayesian approximation to model uncertainty in deep neural networks. They showed that applying dropout before every weight layer in a neural network is mathematically equivalent to a deep Gaussian Process. By performing multiple stochastic forward passes for the same input, one can derive a distribution over outputs, thereby estimating predictive uncertainty.

Building upon this, Miok et al. (2020) applied MC-Dropout to BERT models and Liu et al. (2023) applied it on Conditional Random Field (CRF). They found that the resulting uncertainty estimates were more reliable than traditional deterministic predictions. They noted that BERT, being pre-trained with a dropout rate of approximately 10%, naturally lends itself to this technique. During inference, dropout-induced variability across multiple passes was used to quantify confidence, and the results showed strong potential in applications requiring uncertainty estimation.

There have been some comparative studies by Xiao et al. (2022) and Holm et al. (2023) compared the MC-Dropout strategy with multiple forward passes with direct SoftMax with a single forward pass without dropout to estimate model uncertainty for downstream Classification tasks. They observed that while MC dropout produces the best uncertainty approximations, using a simple SoftMax leads to competitive, and in some cases better, uncertainty estimation for text classification at a much lower computational cost.

## 2 PROBLEM STATEMENT

The objective is to identify and filter incorrect NER predictions without significantly impacting correct extractions. A substantial portion of such erroneous predictions can be mitigated through additional fine-tuning of the NER model using feedback data. However, we observed that beyond a certain point, this approach yields diminishing returns. This limitation arises primarily due to the inherent complexity and variability of real-world clinical data, which is difficult to comprehensively capture during model training. Sole reliance on retraining results in a perpetual catch-up cycle.

The need is to develop a mechanism for detecting and suppressing noisy predictions before they reach downstream pipelines. These noisy predictions typically arise from two sources: (1) Out-of-Distribution (OOD) data points that were not encountered during model fine-tuning, and (2) domain-specific idiosyncrasies inherent in clinical text. Medical practitioners frequently use abbreviations and shorthand notations, the interpretation of which is highly context-dependent. These contextual shifts often lead to ambiguity and misclassification. Moreover due to high volume of text we require a lightweight, adaptable uncertainty estimation mechanism that could dynamically adjust to local context without imposing significant computational overhead.

## 3 BACKGROUND

Consider a BERT NER model taking an input $x \in R^V$, representing an input token vector over vocabulary space $V$, producing final activations $z = \phi(x) \in R^D$, where $D$ is the dimension of hidden layer. This activation $z$ is passed through a SoftMax function to give predictions $\hat{y} = \sigma(z) \in$

$[0, 1]^K$, one-hot-encoded K output NER classes. The SoftMax confidence can be computed as

$$\sigma(\mathbf{z})_i = \frac{\exp(\mathbf{w}_i^\top \mathbf{z})}{\sum_{j=1}^{K} \exp(\mathbf{w}_j^\top \mathbf{z})} \tag{1}$$

where the final layer weight matrix $W \in R^{D \times K}$ can be indexed so that $w_i$ represents column $i$ of the matrix. This transformation converts token embeddings $z$ to $K$ NER classes.

Many uncertainty metrics can be defined from SoftMax, two basic ones are, firstly max probability:

$$U_{max}(z) = max_i \sigma(z)_i \tag{2}$$

Secondly, it can be defined via the entropy:

$$U_{entropy}(z) = -\sum_{i=1}^{K} \sigma(z)_i \log \sigma(z)_i \tag{3}$$

We can't fully rely on SoftMax confidence alone as it exhibits several limitations (described below). So we will later introduce more features to capture uncertainty in a more holistic way.

### 3.1 LIMITATIONS OF SOFTMAX CONFIDENCE FOR UNCERTAINTY ESTIMATION

Pearce et al. (2021) note that in low-dimensional input spaces, SoftMax may tend to nonsensically extrapolate with increased confidence. This is particularly problematic in the context of OOD data, as it is unlikely to contain the distinguishing features that the network was trained upon. Hence OOD activations tend to be of lower magnitude, this results in unusual patterns of final-layer activations.

The following are key failure modes associated with SoftMax-based uncertainty:

1. **Overconfident Extrapolations**: OOD inputs may be mapped to extrapolation regions where SoftMax outputs exhibit high confidence despite the lack of training support.

2. **Aleatoric Uncertainty Conflation**: In scenarios with overlapping class distributions, the model may correctly assign low SoftMax confidence to ambiguous in-distribution inputs. Thus making it harder to detect OOD regions and enlarging the extrapolation regions.

3. **Final-Layer Feature Overlap**: Neural networks are not bijective. Consequently, distinct inputs, including OOD samples, can map to similar final-layer activations. Thus making it harder to distinguish between in-distribution and OOD inputs.

4. **SoftMax Saturation**: Sometimes, SoftMax saturates for a proportion of both training and OOD data and maximum probability ($max_i \sigma(z)_i$) is exactly 1.0. This leads to an inability to create an ordering between samples, thereby degrading OOD detection performance.

Among these, Pearce et al. (2021) emphasize that feature overlap in the final layer plays a more critical role in uncertainty estimation failures than overconfident SoftMax extrapolations alone.

## 4 METHODS

We further consider metrics to determine uncertainty, using the fact that all tokens interact with each other. A token's embedding vector captures the information of its neighborhood via SelfAttention and positional encoding. To understand the same we will refer to transformations in BERT Transformer's inference flow as stated by Vaswani et al. (2017).

### 4.1 BERT TRANSFORMER MODEL

Consider a BERT NER model taking an input $X \in R^{T \times V}$, where all the $T$ tokens are one-hot-encoded to vocabulary space $V$. These vectors are passed through the Transformer architecture (step wise transformation details of Transformer function is presented in Appendix-A 8)

$$Z = Transformer(X) \tag{4}$$

So for an input $X \in R^{T \times V}$, we got an output embedding vector $Z \in R^{T \times D}$. The embedding vector $Z$ is further fed to a classification layer to generate $K$ NER classes as the output $Y \in R^{T \times K}$

$$Y_{logits} = W^C.Z \tag{5}$$

$$Y = SoftMax(Y_{logits}) \tag{6}$$

The semantic vectors of all tokens interact with each other along with their respective positional information. Based on the attention mechanism they influence the final layer embedding vector of other tokens. We will use this property to define a new metric.

## 4.2 DENSITY ANALYSIS IN PROBABILITY SPACE

Equation (6) maps each token to a probability space, where the probability vector is $Y_i \in R^K$, which sums up to 1 and states the likelihood towards one of the $K$ NER classes. As a result of all Transformer operations the embedding vector $Z_i$ generated in Equation (4) contains information about other tokens. As per the transformations in Equation (5) and (6) the probability vector $Y_i$ also contains information about all other tokens.

In the context of NER as per the CoNLL labelling schema for a single entity we will have $K = 3$ (for multiple $E$ entities, $K = 2E + 1$). For simplicity we will illustrate for a single entity with classes: {B-Entity, I-Entity, O}, signifying beginning and intermediate part of entity along with the Other (O) class. In the probability space every token will have a probability vector $Y_i = [prob_B, prob_I, prob_O]$.

We define a $ProbabilityDensityMap$ around the NER predicted token position $t_{predicted}$ to assess its relative strength as per its neighboring tokens. Given probability space is finite $[0, 1]$, we divide the space into $B$ bins (set to 10 in implementation) for each of the $K$ classes (3 for single entity). For predicted token position $t_{predicted}$, we sum the probabilities per class for all neighboring tokens in each bin and weight then as per the token distance $|t - t_{predicted}|$ with a exponential decay function:

$$W_t = e^{\left(\frac{-(|t - t_{predicted}|)^2}{2.R^2}\right)} \tag{7}$$

where, $t$ is token position and $R$ is decay rate, a hyper-parameter for optimal decay effect

---

**Algorithm 1** Function for computing probability density map

1: **function** PROBABILITYDENSITYMAP(B, K, T, Y, $t_{predicted}$)
2:     // B no. of bins, K set of classes, T set of tokens, Y set of output vectors, $t_{predicted}$ token
3:     $numTokens \leftarrow len(T)$
4:     **for** t in T **do**
5:         **for** k in K **do**
6:             **if** $t \neq t_{predicted}$ **then**
7:                 $probability \leftarrow Y[t][k]$
8:                 $bin \leftarrow Integer(probability * B)$
9:                 $W_t \leftarrow exp(-(|t - t_{predicted}|)^2/(2.R^2))$
10:               $Map[bin][k] \leftarrow Map[bin][k] + W_t * probability/numTokens$
    **return** Map

---

## 4.3 CAPTURING TRANSFORMER'S SEMANTIC-PULL VIA PROBABILITYDENSITYMAP

As discussed earlier, during Transformer training, all token embeddings interact with one another through the *Self-Attention* mechanism, which assigns context-dependent weights and adjusts for positional information using positional embedding vectors. When a BERT encoder is fine-tuned for a specific NER task, such as biomarker extraction. The back-propagation process causes tokens corresponding to biomarker entities to receive higher attention weights. This occurs because the model is being optimized to identify such entities. Consequently, words that are semantically or contextually related to biomarkers, particularly those that frequently co-occur near biomarker mentions, acquire elevated weight values in the *Self-Attention matrix* (see Equation 11h, Appendix-A, 8).

This phenomenon leads to what we term a ***Semantic-Pull*** of embedding vectors in the latent representation space for biomarker-related concepts. As indicated in Equation (4) $Z$ denotes a $T \times D$ matrix, where each token $T$ is represented by an embedding vector of dimension $D$. When these

vectors are passed through the *dense linear layer* described in Equation (5), they are transformed into probability vectors $Y$, forming a $T \times K$ matrix as shown in Equation (6). These probability vectors also exhibit the **Semantic-Pull** effect.

To illustrate this concept, consider following sentences and their corresponding NER outputs:

- *Sentence-1*: Patient is treated for ER positive breast carcinoma → [NER-Prediction]: ER
- *Sentence-2*: Patient reported severe chest pain admitted to ER → [NER-Prediction]: ER

In *Sentence-1*, "ER" refers to Estrogen Receptor, a valid biomarker entity, whereas in *Sentence-2*, "ER" denotes Emergency Room, representing a false positive (FP) in the biomarker NER context.

Here is NER's output for *Sentence-1*, generating NER class (B, I, O) probabilities for each token:

| Patient | | | is | | | treated | | | for | | | ER | | | positive | | | breast | | | carcinoma | | |
|---|---|---|---|---|---|---|---|---|---|---|---|---|---|---|---|---|---|---|---|---|---|---|---|
| B | I | O | B | I | O | B | I | O | B | I | O | B | I | O | **B** | **I** | O | B | **I** | O | B | **I** | O |
| 0. | 0. | 1. | 0. | 0. | 1. | 0. | 0. | 1. | 0. | 0. | 1. | 1. | 0. | 0. | **.001** | **.048** | .951 | 0. | **.003** | .997 | 0. | **.002** | .998 |

Here is NER's output for *Sentence-2*, generating NER class (B, I, O) probabilities for each token:

| Patient | | | reported | | | severe | | | chest | | | pain | | | admitted | | | to | | | ER | | |
|---|---|---|---|---|---|---|---|---|---|---|---|---|---|---|---|---|---|---|---|---|---|---|---|
| B | I | O | B | I | O | B | I | O | B | I | O | B | I | O | B | I | O | B | I | O | B | I | O |
| 0. | 0. | 1. | 0. | 0. | 1. | 0. | 0. | 1. | 0. | 0. | 1. | 0. | 0. | 1. | 0. | 0. | 1. | 0. | 0. | 1. | .999 | 0. | .001 |

Examining the NER output for *Sentence-1*, we observe that, in addition to the true biomarker token "ER", nearby tokens such as "positive", which denotes the biomarker test result, exhibit non-zero probabilities for the B-Biomarker and I-Biomarker classes. Furthermore, because the "ER" biomarker is typically associated with breast cancer diagnoses, the tokens "breast carcinoma" also display non-zero probabilities for the I-Biomarker class (highlighted in bold in above table).

In contrast, the NER output for *Sentence-2* shows no tokens besides "ER" with non-zero probabilities for either B-Biomarker or I-Biomarker classes. This distinction highlights how contextual token neighborhoods yield distinct probability distributions between true and false entity cases.

Next, we describe the binning strategy used for probability analysis. We first present the intermediate cumulative probability outputs per bin for both sentences to enhance interpretability. This is followed by the Probability Density Map (PDM) generation, as detailed in Algorithm-1 (1), for each bin corresponding to both sentences. Since all probability values fall either below 0.1 or above 0.9, only Bin-1 and Bin-10 contain non-empty values, remaining bins are therefore omitted below.

*Sentence-1*: Cumulative probability bins

| Bin-1 | | | Bin-... | | | Bin-10 | | |
|---|---|---|---|---|---|---|---|---|
| **B** | **I** | O | B | I | O | B | I | O |
| **0.001** | **0.053** | 0.0 | 0.0 | 0.0 | 0.0 | 0.0 | 0.0 | 6.946 |

*Sentence-1*: **Probability Density Map (PDM)**

| Bin-1 | | | Bin-... | | | Bin-10 | | |
|---|---|---|---|---|---|---|---|---|
| B | **I** | O | B | I | O | B | I | O |
| 0.0 | **0.004** | 0.0 | 0.0 | 0.0 | 0.0 | 0.0 | 0.0 | 0.185 |

*Sentence-2*: Cumulative probability bins

| Bin-1 | | | Bin-... | | | Bin-10 | | |
|---|---|---|---|---|---|---|---|---|
| B | I | O | B | I | O | B | I | O |
| 0.0 | 0.0 | 0.0 | 0.0 | 0.0 | 0.0 | 0.0 | 0.0 | 6.999 |

*Sentence-2*: **Probability Density Map (PDM)**

| Bin-1 | | | Bin-... | | | Bin-10 | | |
|---|---|---|---|---|---|---|---|---|
| B | I | O | B | I | O | B | I | O |
| 0.0 | 0.0 | 0.0 | 0.0 | 0.0 | 0.0 | 0.0 | 0.0 | 0.094 |

As we can see the PDM of *Sentence-1* with true Biomarker has non-zero values in Bin-1, while the same is not true for *Sentence-2*. Hence, these PDM bins can act as good distinguishable features that can be used by the Noise Removal (NR) classifier to identify noisy Biomarker entities.

### 4.4 Embedding space vs Probability space correlation

The $D$-dimensional embedding vectors $Z$ (Equation 4) were projected into a two-dimensional space using Multi-Dimensional Scaling (MDS), a manifold learning technique that preserves global geometric relationships within the embedding space. Consequently, tokens that are distant in high-dimensional space remain distant after projection, and the converse holds as well.

Probability-space visualizations were created by directly plotting the output probability vectors $Y$ (Equation 6) in a three-dimensional coordinate system, where each axis corresponds to one of the NER classes {B, I, O}.

In these plots, the TP entity token is represented by a green star, and the FP token by a red star. Neighboring tokens of TP entities are shown as green upward triangles, while those of FP entities are denoted by red downward triangles. This convention is consistently applied in both the embedding and probability space plots. For additional contextual reference in embedding space, blue circles

represent unrelated entities, light green circles denote correctly identified tokens of the same entity class, and pink circles indicate false positives within the same class.

The analysis uses the same two example sentences from Section 4.3, where the token "ER" appears in distinct clinical contexts, *Sentence-1* as a TP and *Sentence-2* as a FP:

- *Sentence-1*: Patient is treated for ER positive breast carcinoma → [NER-Prediction]: ER

- *Sentence-2*: Patient reported severe chest pain admitted to ER → [NER-Prediction]: ER

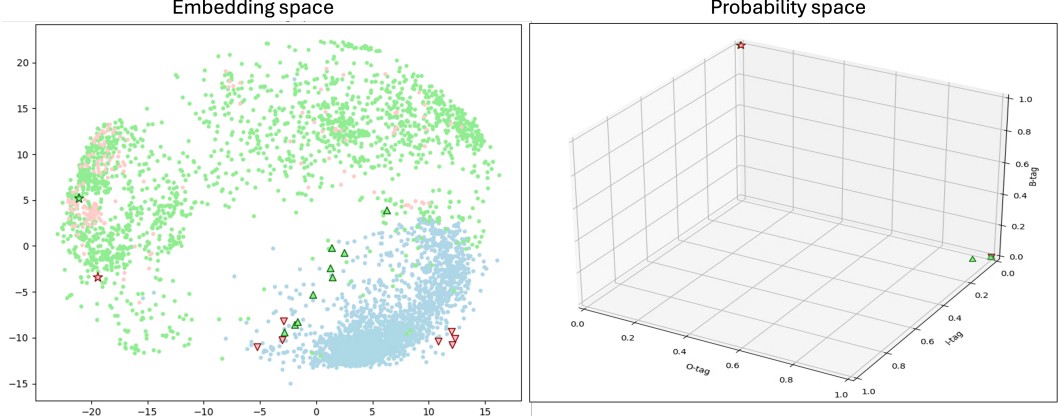

Figure 1: Left plot shows a 2-dimensional manifold of embeddings from Biomarker NER model, with stars indicating predicted entity, TP as green and FP as red, triangles representing neighborhood. The space depicts training data, where green circles indicates Biomarkers and pink as FPs. Right plot indicates the predicted entity and its context token in the probability space.

**Embedding Space Analysis:** As illustrated in Figure 1, the true positive (TP) token "ER" (green star) lies within a region densely populated by correctly identified biomarker entities (light green circles). In contrast, the false positive (FP) "ER" (red star) occupies a sparse region near the TP cluster boundary, indicating partial semantic overlap. This proximity suggests that the NER model struggles to fully disambiguate contextually similar entities.

Examining the neighboring tokens (triangles) further reveals that red triangles representing FP neighborhoods tend to lie within or adjacent to the blue region corresponding to other entity classes. Interestingly, the green triangles, denoting TP neighborhoods, are also close to the blue region but exhibit a distinct **semantic-pull** toward the green space, with several surrounded by green circles that signify higher semantic coherence.

**Probability Space Analysis:** In the corresponding probability space (Figure 1), both "ER" tokens occupy the upper-right quadrant, reflecting high B-tag confidence. The overlap of red and green stars indicates that both were confidently predicted as biomarker onsets. However, contextual differences emerge upon closer inspection: for *Sentence-1*, the neighboring tokens (green triangles) largely align with the O-tag high-probability range, with some exhibiting mild I-tag activation, consistent with semantically related terms like positive. This reinforces the **semantic-pull** effect, wherein frequent co-occurrence in training data induces probabilistic affinity toward biomarker classes.

Conversely, in *Sentence-2*, neighboring tokens (red triangles) are positioned near the blue region in the embedding space and close to the O-tag axis in probability space, reflecting the absence of biomarker-related context.

These observations validate our hypothesis that contextual and density-based features derived from probability distributions can effectively capture uncertainty and help flag ambiguous NER predictions.

## 5 MODEL FEATURES

We have used two kinds of features, density based and statistical features. The most important feature - PDM (Probability Density Map), have been defined in detail in Section 4.2. We will now introduce statistical features to capture more forms of uncertainty from NER's probability space.

*Density features*: Represented as 10 bins map per NER class

$$PDM_{binsArray} = ProbabilityDensityMap(PredictedTokenNeighbourhood)$$

*Statistical features*: All statistical operators are applied on all operands like a cross product

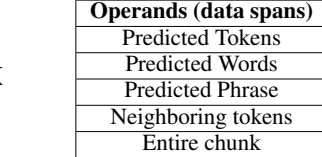

| Statistical operators |
|---|
| Mean, Max |
| Coeff. of Variation |
| Entropy |
| Probability ratios |
| Probability deltas |

**X**

| Operands (data spans) |
|---|
| Predicted Tokens |
| Predicted Words |
| Predicted Phrase |
| Neighboring tokens |
| Entire chunk |

The complete list of features with details is available in Appendix-B 8 with definitions.

**Model explainability**: Since DecisionTree is an explainable model, so we have added an example of Decision path from the DecisionTree model in Appendix-C 8, where for a sample text we showcase how these features identify a noise from a true concept.

## 6 SOLUTION AND DATA OVERVIEW

To assess the impact of epistemic uncertainty arising from OOD inputs and misclassification errors, along with impact of SoftMax extrapolations. We conducted a focused analysis using NER models trained on clinical entities. A diverse set of clinical documents was sampled, and annotated by business scripts and manually reviewed by clinical subject matter experts (SMEs), resulting in labeled sets of True Positives (TPs) and False Positives (FPs). Here the FPs were meticulously chosen by SMEs, which were lexically similar but were irrelevant when considered with neighbouring context.

We chose two datasets for testing these models:

- EMR: Real clinical dataset of 50K Lung & Breast cancer patients treated in 2024

- MIMIC-III: Publicly available anonymized clinical data of 40K patients with 2 million docs

EMR: Entity data distribution

| Data type \ Entities | Biomarker | Tumor type | Surgery | Medication | Tumor Grade | Histology |
|---|---|---|---|---|---|---|
| True clinical entities | 6.7K | 1.7K | 1.6K | 10.6K | 1.5K | 1.4K |
| Similar false entities | 24.5K | 1.4K | 10.9K | 46.7K | 0.5K | 19.8K |

MIMIC data distribution

| Data type \ Entities | Biomarker | Tumor type | Surgery | Medication |
|---|---|---|---|---|
| True clinical entities | 200 | 500 | 950 | 5,6K |
| Similar false entities | 150 | 100 | 100 | 1K |

Using features mentioned in section 5, we trained a supervised Decision Tree classifier to categorize each NER prediction into one of two classes: {Strong, Weak}. During training, SME-labeled TPs were assigned to the Strong class, whereas FPs were assigned to the Weak class. At inference time, each NER prediction token, along with its corresponding feature set, was passed to the Decision Tree classifier, which produced a binary classification outcome {Strong, Weak}. Predictions classified as *Weak* were treated as potential noise terms. Importantly, the Decision Tree, being an interpretable model, also provides explicit decision paths, enabling SMEs to examine the rationale for why a particular clinical entity was identified as noise.

# 7 RESULTS

To systematically evaluate the performance of the proposed Decision Tree-based Noise Removal (NR) models, we selected multiple clinical entities that are critical to clinical trial workflows, each requiring both high precision and high recall. As a baseline, we fine-tuned BERT-based NER models on EMR data, achieving entity-level F1-scores close to 0.9.

We then compared the F1-scores of the baseline NER model against widely used approaches for reducing False Positives (FPs). For each method, hyper-parameters were optimized to identify the best-performing configuration. Specifically, the optimal threshold value was selected for SoftMax Thresholding, the best temperature parameter $T$ was chosen for Temperature Scaling, and the most effective mean–variance cutoffs were determined for Monte Carlo (MC) Dropout. These methods were compared against our NR approach, which applies a supervised Decision Tree model to the baseline NER predictions using engineered features derived from the model outputs.

Table 1: **F1-score** comparison on **EMR** data over different clinical entity types

| Element | Base NER | SoftMax Thresholding | Temp. Scaling | MC Dropout | NER + NR |
|---|---|---|---|---|---|
| Biomarkers | 0.933 | 0.935 | 0.935 | 0.935 | **0.953** |
| Surgery | 0.927 | 0.904 | 0.889 | 0.903 | **0.942** |
| Drug name | 0.895 | 0.906 | 0.913 | 0.919 | **0.932** |
| Tumor Grade | 0.914 | 0.909 | 0.900 | 0.905 | **0.925** |
| Tumor | 0.877 | 0.880 | 0.882 | 0.879 | **0.906** |
| Histology | 0.932 | 0.940 | 0.940 | **0.950** | 0.935 |

To further validate our findings, we replicated the experiments on the publicly available MIMIC-III dataset. The models trained on EMR data were applied directly to MIMIC-III without fine-tuning. A subset of MIMIC-III data was randomly sampled and annotated by SMEs for evaluation. Since ground-truth coverage of MIMIC-III was incomplete, we performed a relative comparison across approaches, measuring the percentage reduction in both FPs and TPs relative to the base NER model.

Table 2: **MIMIC-III** Relative comparison from base NER as **(%TP Drop, %FP Drop)**

| Element | SoftMax Thresholding | Temp. Scaling | MC Dropout | NER + NR |
|---|---|---|---|---|
| Biomarkers | (6%, 0%) | (6%, 1%) | (6%, 2%) | **(6%, 88%)** |
| Surgery | (1%, 32%) | (1%, 42%) | **(3%, 68%)** | (2%, 55%) |
| Drug name | (2%, 24%) | (2%, 31%) | (5%, 44%) | **(1%, 47%)** |
| Tumor | (0%, 41%) | (1%, 77%) | (1%, 77%) | **(2%, 87%)** |

The results consistently demonstrated the effectiveness of the NR models. On EMR data, NR achieved the best performance in 5 out of 6 entity categories, and on MIMIC-III data, in 3 out of 4 categories. Notably, on MIMIC-III, the NR models exhibited a conservative behavior, prioritizing maximal FP reduction while constraining TP loss. We set the hyper-parameter to limit TP drop to no more than 5–6%, while allowing FP reductions to be maximized. Under these conditions, NR reduced noise (FPs) by 47% to 87% while keeping TP degradation below 6%. This demonstrates that NR can substantially increase precision with minimal loss in recall.

# 8 CONCLUSION

We proposed a lightweight, explainable post-processing approach for enhancing the precision of NER outputs in clinical text, particularly in the presence of out-of-distribution data. By leveraging internal probabilistic signals and embedding space characteristics, we engineered statistical and density features that feed into a Decision Tree-based Noise Removal (NR) model.

The resulting NR models significantly improved precision across clinical entities with minimal recall degradation. Crucially, this method operates without any modifications to the original NER training or inference processes, making it a practical and efficient solution for real-world clinical deployments requiring high precision and robustness.

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

## APPENDIX A: TRANSFORMER EQUATIONS

A BERT NER model taking an input $X \in R^{T \times V}$, for $T$ tokens in input that are one-hot-encoded to vocabulary space $V$. These one-hot-encoded vector X goes through the following steps to generate the Transformer output, as vectors in embedding space of dimension $D$.

$$X_{E_w} = W^E.X \tag{8}$$

$$X_{E_{pos}} = \begin{cases} sin(pos/10000^{2i/D}) & \text{if i is even, } \forall \text{ pos } \in [0, \text{T}), \forall \text{ i } \in [0, \text{D/2}) \\ cos(pos/10000^{2i/D}) & \text{if i is odd, } \forall \text{ pos } \in [0, \text{T}), \forall \text{ i } \in [0, \text{D/2}) \end{cases} \tag{9}$$

$$X_{pos} = X_{E_w} + X_{E_{pos}} \tag{10}$$

$$Z_{attn} = MultiHeadAttention(X_{pos}, X_{pos}, X_{pos}) \tag{11a}$$

$$\text{where, } MultiHeadAttention(Q, K, V) = Concat(Head_1, ..., Head_h).W^O \tag{11b}$$

$$\text{where, } Head_i = Attention(Q_{head_i}, K_{head_i}, V_{head_i}) \tag{11c}$$

$$\text{where, } Q_{head_i} = Q.W_i^Q \tag{11d}$$

$$\text{where, } K_{head_i} = K.W_i^K \tag{11e}$$

$$\text{where, } V_{head_i} = V.W_i^V \tag{11f}$$

$$\text{where, } Attention(Q, K, V) = AttentionWeights(Q, K).V \tag{11g}$$

$$\text{where, } AttentionWeights(Q, K) = SoftMax(\frac{Q.K^T}{\sqrt{D}}) \tag{11h}$$

$$Z_{norm} = LayerNorm(X_{pos} + Z_{attn}) \tag{12}$$

$$Z_{ffn} = FFN(Z_{norm}) \tag{13a}$$

$$\text{where, } FFN(x) = max(0, x.W_1 + b_1).W_2 + b_2 \tag{13b}$$

$$Z = LayerNorm(Z_{ffn} + Z_{norm}) \tag{14}$$

The final output of Transformer is an output embedding vector $Z \in R^{T \times D}$. That can be used for multiple purposes, one directly as a semantic vector to represent the text concept. Or can be fed to subsequent linear layers for further classification purpose like SentenceClassification, NER (Named Entity Recognition), RE (Relation Extraction), ABSA (Aspect Based Sentiment Analysis), etc.

## APPENDIX B: FEATURE DEFINITIONS

The proposed features are designed to extract and represent nuanced information from the final output layer of a BERT-based NER model. As described in a preceding sections, the probability space offers a rich reservoir of information, which includes both the intra-class probability strength of individual tokens and their interdependencies with surrounding tokens.

Feature extraction is conducted across multiple levels of textual granularity. The following contexts are considered for generating features:

- Predicted Token: The most granular unit predicted by the BERT model as a (B, I) class
- Predicted Word: A group of predicted tokens with (B, I) class constituting a complete word
- Predicted Phrase: A sequence of words forming a coherent phrase, with predicted (B, I) classess
- Immediate Neighboring Tokens: Tokens that are immediately adjacent (preceding or following) to the predicted phrase belonging to any class
- Entire Context: All tokens in the sentence excluding the predicted phrase.

Two primary types of features are computed:

- Statistical Summary Features
- Spatial Density Features

**Statistical Summary Features:** These features are computed for the predicted token, word, or phrase. They capture aggregate statistical characteristics of the predicted probability distributions. Specifically, the following metrics are extracted:

- Class Counts: Frequency of each predicted class within the group.
- Class Ratios: Proportional representation of each class.
- Maximum Probability: The highest class probability among tokens.
- Mean Probability: Average probability across tokens.
- Coefficient of Variation: Standard deviation normalized by the mean, indicating the spread of the probabilities.

- Probability Difference: Difference between the highest and second-highest class probabilities.

- Probability Ratio: Ratio of the highest to the second-highest class probabilities.

**Spatial Density Features:** These features are computed based on a discretized view of the probability space:

- Each class label (B, I, O) is assigned ten bins based on probability values.

- Each bin has a width of 0.1, covering the range (0, 1).

- Tokens are mapped into one of 30 total bins (10 per class) based on their predicted probability and class label.

- Tokens are further weighted by their distance from the predicted token using a Gaussian decay function:

$$W = \exp^{(\frac{-x^2}{2R^2})}$$

  - where $x$ is the relative distance from the predicted token and $R$ is a hyper parameter controlling the decay rate.

These spatial density features are designed to capture the structural distribution of class probabilities within the context of the predicted token, taking into account the spatial arrangement of probabilities across the sequence.

## APPENDIX C: MODEL INFERENCING EXPLANATION

The nomenclature of features abide a convention, where multiple sub-words separated by '_'. The first sub-word can have one of the following values: (SPD, Context, Phrase, Word, Token), where SPD indicates this is a Spatial Density feature, while the other words indicate it is a statistical feature with context window as one of (Context, Phrase, Word, Token). The second sub-word indicates the NER class type can have one of these values: (B-tag, I-tag, O-tag). Then third and fourth sub-tokens indicates the type of statistical measure used.

To illustrate model interpretability, we present a synthetic example: *Sample text:* "Signed Final report Impression / Plan patient with no PMH, 14 pack years, stage IVb cT1N2M1c adenocarcinoma of the lung (RLL primary met to the brain, PDL1 50% and positive ALK on Foundation ctDNA), ALK FISH negative at Stanford with +ALK noted on Stanford STAMP testing."

In the example above, in addition to correctly identified biomarkers such as ALK and PDL1, the NER model erroneously predicted the word "met" as a biomarker. While "MET" is indeed a known biomarker, in this context, the word "met" denotes metastasis to the brain, not the gene. Despite being assigned a high probability as a B-Biomarker token by the NER model, the Decision Tree correctly identified this as a weak candidate and recommended it for removal.

Due to the explainable nature of Decision Trees, we extracted the decision path used to reach this conclusion:

```
Decision Path:  (SPD_B-tag_WCount_bkt_0.9-1.0 >
7.2e-7) & (Token_prob_class_ratio_3_by_2 <= 0.667) &
(SPD_O-tag_WCount_bkt_0.9-1.0 <= 7.7) & (Phrase_prob_O-tag_ratio
<= 0.7) & (SPD_O-tag_WCount_bkt_0.9-1.0 <= 6.696) &
(Token_prob_class_ratio_3_by_2 <= 0.653) & (Word_B_I-tag_count
<= 3.5) & (Context_B-tag_mean_prob > 3.18e-6) &
(Token_prev_word_max_prob > 0.944)
```

This decision path highlights how the model utilized spatial density (SPD) features, focusing on the distribution of high-confidence predictions across context windows. It also leveraged the ratio of third-to-second class probabilities as an uncertainty metric and evaluated phrase-level coverage of the 'O' class to quantify contextual alignment. Since "met" was the only predicted token and formed a single-token phrase, hence all the token, word and phrase features pertained solely for the "met" token. Additionally, the model considered the probability strength of adjacent tokens to assess semantic consistency.

An SME evaluating the isolated token "met" might initially consider it a biomarker. However, contextual cues, especially references to brain metastasis, clearly disqualify it. The Decision Tree model replicated this human reasoning by prioritizing local contextual signals, effectively down-weighting surrounding biomarkers and correctly identifying the term as a false positive.

