# OpenReview forum: "Noise reduction in BERT NER SLM models for clinical entity extraction in clinical trials"
_ICLR.cc/2026/Conference — Submitted to ICLR 2026_

### Official Review · Reviewer_acuM · 2025-10-23

**Soundness:** 2
**Presentation:** 2
**Contribution:** 2
**Rating:** 2
**Confidence:** 4

**Summary:**

The authors propose to train a decision tree over various features as a form of confidence calibration over BERT models. The approach is evaluated on two medical named entity recognition datasets.

**Strengths:**

* There is a clear need for continued work on confidence calibration.
* The authors compare to some common calibration techniques: temperature scaling (2017) and MC-Dropout (2016)

**Weaknesses:**

Major:
* The authors do not compare to confidence calibration algorithms newer than 2017, e.g., SoftmaxCorr (2024) in the related work, RelCal (2023) https://link.springer.com/chapter/10.1007/978-3-032-05962-8_14, etc. Given the complexity of the proposed feature set for the proposed decision tree classifier, comparisons to such recent work are essential.
* The evaluation datasets are not well defined. No sizes are given. The main dataset is defined only as "EMR data", with no reference for the dataset, or if the dataset was created internally, with no information about the annotation process (e.g., agreement). The MIMIC-3 dataset has no information about inter-annotator agreement.
* Even after looking at appendix A and appendix C, the exact feature definitions are unclear to me.  Features need to be defined formally, with a brief example to demonstrate each feature.
* The paper spends the far too much space (roughly the first four pages) arguing in different ways that confidence calibration is important, and the presentation of the key contribution doesn't show up until Figure 1 on page 7. That content needs to be dramatically compressed, probably down to a maximum of 2 pages so that the key contribution shows up no later than page 3, and so there is room to address the other weaknesses noted here.

Minor:
* Calling BERT a small language model seems odd, given that it's millions of parameters. Consider using a different term.
* This sentence suggests that there are only two important metrics: "Two basic uncertainty metrics can be defined from SoftMax" Consider either justifying the choice of only those two, or include other well-known metrics like margin sampling (best vs. second best), least confident, etc. https://burrsettles.com/pub/settles.activelearning.pdf
* Equation (9) is incorrect; BERT uses positional embeddings, not positional sinusoidal encodings. Consider abbreviating equations 8-18, which are just repeating the standard transformer definition, with something like Z = Transformer(X).

**Questions:**

* Why K = 3 for CoNLL? Shouldn't it be larger than that since there are E>1 entity types in CoNLL, so K = 2 * E + 1 > 3?
* Could you explain a bit more the intuition behind ProbabilityDensityMap? What is it trying to achieve and why would we expect that to be a good idea?
* Are there triangles (mentioned in the text) in Figure 1? I don't see any.
* Will the annotated subset of MIMIC-3 be released?

---

> ### Author Response · Authors · 2025-11-24
>
> First of all thanks for reviewing our paper and providing your valuable feedback.
>
> We will begin with addressing the major concerns raised for the paper:
> * Regarding other methods (e.g., SoftMaxCorr): As noted in Section 1.1 (Related Works), we did not include SoftMaxCorr or RelCal in our result comparisons for specific reasons. SoftMaxCorr operates at a batch level by comparing correlation matrices between reference and inference phases, whereas clinical-grade models require record-level noise detection. Since our NER models already achieve F1 scores in the high 80s to low 90s, the NR models aim to further improve precision into the high 90s. Given that only a small fraction of millions of processed records contain noise, batch-level correlation methods are ineffective for such fine-grained detection.
>     * On calibration methods: Our paper focuses on noise removal (reducing false positives) rather than probability calibration. Transformer architectures are complex and require substantial pre-training data, and no existing calibration methods have demonstrated consistent success in both performance and probability calibration. Therefore, we focused on approaches that operate post-training without modifying the Transformer architecture. Since RelCal alters the training process, it was not evaluated.
> * Dataset details: We acknowledge the omission and will include them in the revised version. For EMR data, we can share only the cohort selection criteria and patient counts. For MIMIC-III, we can release the annotated dataset used for evaluation. All annotations were performed by internal SMEs, and inter-annotator agreement metrics can be shared if required.
> * Feature definitions: We will formally include all feature definitions in the main paper in the revised version.
> * Uncertainty metrics: In addition to entropy and maximum probability, we computed several SoftMax-based features, such as probability deltas between the top classes (1st–2nd, 2nd–3rd), then probability ratios among classes (1st–2nd, 2nd–3rd), as well as statistical measures like mean, coefficient of variation, and maximum. These are currently listed in the Appendix, and we will rephrase them more clearly in the revision.
> * Paper organization: We will condense other sections to accommodate detailed feature definitions within the main paper.
>
> We will now try to address the questions raised:
> Answer-1:
> We developed multiple NER models, some designed to extract multiple entities and others focused on complex single entities. For example, Biomarker extraction, being more complex, uses a single NER model with three classes (B-Biomarker, I-Biomarker, O). In contrast, entities like Stage and Grade are handled by a combined model with five classes (B-Stage, I-Stage, B-Grade, I-Grade, O).
> Thus, for single-entity NER models, K = 3, while for multi-entity models, K = 2E + 1 (> 3), where E is the number of entity types.
>
> Answer-2:
> As discussed in Section 5.1 (Feature Motivation), there is a strong relationship between the semantic embedding space and the probability space. In the embedding space, true biomarker terms are surrounded by semantically similar concepts, whereas false biomarkers have unrelated neighbors. This pattern is mirrored in the probability space, where neighboring tokens of true biomarkers exhibit higher likelihoods of being biomarkers, as shown in the right-hand probability plot of Figure 1.
>
> The Probability Density Map captures a snapshot of this distribution for a text segment. Since probabilities lie within [0,1], we discretize them into ten bins of width 0.1 and record the neighboring token distribution as a ten-bin signature. This signature differs between true and false biomarkers, allowing a Decision Tree classifier, used in the NR (Noise Removal) models, to effectively distinguish between them.
>
> Answer-3:
> As described in Section 5.1, triangles represent the mean-pooled vector of all tokens comprising a multi-token entity. For example, the word EGFR is tokenized into “EG” and “FR” by tokenizer, producing two embedding vectors that are averaged to form a single word vector. Tokens are depicted as stars; hence, in this case, two stars and one triangle will be shown. A similar multi-token example is provided in Appendix B, Case 1, where Estrogen Receptor is split into three tokens: [“Est”, “rogen”, “Receptor”].
>
> Conversely, if a word is represented by a single token, such as met, mean pooling is unnecessary. Therefore, in Figure 1, such cases are represented only by a star and not by a triangle. That is why there are no triangle in Figure-1, but are present in Figure-2 in Appendix-B.
>
> Answer-4:
> Yes, we can release the annotated subset of the MIMIC-III dataset that we labeled specifically for evaluating the NR models, if required.

---

> > ### Comment · Reviewer_acuM · 2025-11-24
> >
> > * Thanks, I now understand that you're only considering calibration methods that are post-training. That seems reasonable. But I believe there are more recent papers that meet your criteria, such as https://arxiv.org/abs/2302.05118, https://ojs.aaai.org/index.php/AAAI/article/view/34120, or the "5.2 Soft Calibration Objectives for Post-Hoc Calibration" section of https://proceedings.neurips.cc/paper/2021/hash/f8905bd3df64ace64a68e154ba72f24c-Abstract.html. While your goal may not be calibration, the techniques you are using as baselines are calibration techniques, so it's important to compare against the most recent such techniques.
> > * Thank you for committing to include more dataset details, improve feature definitions, rephrase the uncertainty metric claim, and condense introductory sections. However, I am unable to judge the adequacy of those revisions without seeing them.
> > * Thanks for your answers to my questions. They have resolved my uncertainties for those portions of the paper. Please do include some version of your answers in your revised main text.

---

> > > ### Author Response · Authors · 2025-11-25
> > >
> > > Thanks for the additional feedback, we will try to upload a revised version soon, so that you can take a look at the changes suggested.

---

> > > > ### Author Response · Authors · 2025-12-01
> > > >
> > > > We have uploaded the revised version of the paper, with the following changes:
> > > > * We have updated the Abstract and Introduction to elaborate the high level approach
> > > > * We have also added more theory to explain the Semantic-Pull effect that our NR models are capturing
> > > > * We have added a section to elaborate the importance of Probability Density Map
> > > >     * We have added example and showcased how the Probability Density Map captures the snapshot of neighboring tokens
> > > >     * We have re-written the Probability Density Map formula and equation to make it more comprehendible
> > > > * We have updated the Embedding space and Probability space plots with more details, showing both predicted entity (star) and neighboring tokens (triangles)
> > > > * We have compressed most of the Introduction section and also compressed the Transformer equations to add more details about our methods
> > > > * We have re-written the features section for better readability
> > > > * We have details on the data distributions
> > > > * Renamed and re-positioned a few sections to better organize the content

---

### Official Review · Reviewer_eD5m · 2025-10-31

**Soundness:** 2
**Presentation:** 2
**Contribution:** 2
**Rating:** 2
**Confidence:** 4

**Summary:**

This paper aims to improve the precision of BERT-based small language models for clinical entity extraction task. A simple way is to directly filter out predictions of low probability. However, due to the limitation of Softmax and noise in the training process, this method cannot effectively filter out wrong predictions. This paper proposes a method to distinguish strong and weak predictions by constructing semantic and statistical features for each token position, and training a interpretable decision tree model to classify predictions. This method is lightweight and achieve consistent improvement compared with other baseline prediction filtering methods on EMR and MIMIC-III data.

**Strengths:**

1. The proposed method can be operated without any modifications to the original NER method or training process, which makes it practical.

2. The main paper gives a detailed explanation in the related work and problem statement.

**Weaknesses:**

1. Feature motivation. The paper used a case study containing two example clinical texts to illustrate the feature motivation, which is not a persuasive and reasonable way.

2. Experiments. The paper does not give a comprehensive experiment to prove the model, including more comprehensive comparison methods and metrics. The experiment section could be expanded to provide more details of the datasets used, and the inference time for each method may be included to prove the efficiency of NER + NR. Also, a simple ablation study for feature construction may further improve the solidity of the experiments.

3. Paper writing. The paper does not arrange the length and content of its paragraphs appropriately. For example, the results and conclusion sections are too brief, providing neither a clear description of the data used nor an adequate explanation of the comparative methods.

**Questions:**

1. What do you think are the advantages and practical values of your method compared with deep learning-based and LLM-based approaches in real-world clinical NER?

2. Should a discussion on the interpretability of the decision tree-based NR model be added to the main paper? Why choose exactly these features for decision tree? Though motivation of feature selection is provided in the appendix, this only provides a general idea such as the model should focus on nearby tokens, but not specific feature design.

3. For the experiment section, how are the hyperparameters of baseline methods selected? Are they tuned on the same training dataset as NER + NR?

4. In Table 2, For the biomarkers element, why other methods have almost no FP drop while NER + NR achieve 88% FP drop? This result seems a bit extreme.

---

> ### Author Response · Authors · 2025-11-24
>
> First of all thanks for reviewing our paper and providing your valuable feedback.
>
> We will begin with addressing the concerns raised for the paper:
> * We will re-articulate the feature definitions more formally to emphasize their motivation and relevance. The key idea is that, without modifying the Transformer model, analyzing the probability space of token-level class predictions enables the detection of various issues, such as out-of-distribution (OOD) patterns, probability mis-calibrations, and data ambiguities arising from overlapping semantic spaces. These observations motivated the creation of multiple statistical and density-based probability features.
> * The model was evaluated on real-world oncology data, which is inherently complex and ambiguous. We replicated the results on the publicly available MIMIC-III dataset for validation. We can share the annotated MIMIC-III dataset and evaluation metrics if required to showcase the models ability to handle complex cases. Our comparisons focus on post-processing methods that do not alter Transformer training. We will include dataset details aling with inference time benchmark in the revised version and plan a feature ablation study to demonstrate the contribution of each feature.
> * Paper organization: We will condense other sections to accommodate detailed feature definitions and comparative method discussions within the main paper.
>
> We will now try to address the questions raised:
> Answer-1:
> As noted in our response to Reviewer 1’s first question, our pipeline integrates three types of models: Custom-ClinicalBERT, the proposed Noise Removal (NR) models, and LLMs such as Llama, each contributing to optimal system performance.
>
> LLMs like Llama exhibit strong higher-order reasoning capabilities but tend to hallucinate during large-scale entity extraction; hence, they are primarily used for higher order inferential tasks. Custom-ClinicalBERT models, while effective and well-controlled for entity extraction with strong F1 scores, are constrained by the variability present in their training data and may not capture all real-world complexities. The NR models address this limitation by filtering out out-of-distribution (OOD) prediction errors, thereby improving overall precision.
> Together, these three model types ensure a robust, high-accuracy, medical-grade information extraction pipeline.
>
>
> Answer-2:
> Yes, we developed an interpretability component for the Decision Tree model, where each root-to-leaf path represents the reasoning behind classifying a sample as noise. Due to space constraints, this analysis could not be included in the main paper but is briefly described in Appendix C.
>
> Our feature motivation study in section 5.1 revealed that the probability space contains rich information, warranting multiple approaches to capture it. Accordingly, we experimented with various statistical and density-based features, which yielded strong performance. The current feature set was selected based on extensive experimentation and feature importance analysis. The core idea behind these features is that probabilities provide good indication of class likelihood, but they are also flawed due to SoftMax limitations explained in section 3.1, hence pure statistical analysis of probabilities may not help, moreover we showed strong correlation between embedding space and probability space in section 5.1. So, neighborhood analysis is allowing us to model the semantic features of Transformers embedding. Thus, by combining both we are getting best of both and able to complement the shortcomings of either of systems on their own.
>
>
> Answer-3:
> In the EHR domain, we have access to a large volume of annotated data curated by our SMEs. Accordingly, we maintained separate test and validation sets for hyper-parameter tuning. Given that our models are lightweight, we employed the GridSearchCV method to efficiently explore the hyper-parameter space and identify optimal configurations.
>
>
> Answer-4:
> In the Results section, Table 1 presents performance on EHR data, while Table 2 reports results on MIMIC-III data. Due to strict data-sharing constraints in clinical research, we included MIMIC-III results as it is a publicly available dataset that allows independent validation. However, MIMIC-III represents general medical data rather than oncology-specific records. It contains substantial information on procedures, medications, and diagnoses, but oncology-specific biomarkers are relatively rare.
>
> Consequently, the number of biomarker instances in MIMIC-III is small, leading to large percentage variations even with minor absolute differences. Moreover, our NER model performs well overall, but on MIMIC-III it is making a few recurring misclassifications. These recurring patterns are missed by other approaches but are effectively corrected by the NR model. These two factors explain why prior methods show minimal percentage improvement, whereas the NR model demonstrates an 88% drop in error rate.

---

### Official Review · Reviewer_UJjB · 2025-11-01

**Soundness:** 3
**Presentation:** 2
**Contribution:** 3
**Rating:** 6
**Confidence:** 3

**Summary:**

This paper presents an analysis of the overconfidence problem in Medical domain adapted pretrained LM-based NER, and engineered features based on the analysis to add a noise reduction model to reduce the false positive predictions.
Contributions
1. Analysis of overconfidence problem in medical NER
2. a well motivated & inexpensive solution with significant improvement

**Strengths:**

1. The feature design was well motivated from the probability space analysis
2. The proposed solution works well for reducing false positives

**Weaknesses:**

presentation could be improved. I had to re-read the analysis section 4.2 and 5.1 several times to understand what the authors were trying to say.
1. eq 21 the notation is a mess -- what does |t_anchor-t| mean? I could roughly infer from appendix but it is confusing as hell
2. "feature used" section was confusing -- is 'cross-product' supposed to mean 'cartesian-product'? I had to read the appendix to clarify how it is actually used
2. better use of space --  the authors spend almost a whole page writing out equations for transformers, and probably resulted in needing to cut out some content to fit in the page limits. I would suggest some parts can be assumed common knowledge or try to compress the space it takes up with some formatting, so you have more room to clarify e.g. your feature design?
3. orders -- I see probability density maps & binning in section 4.2 density analysis, but did not seem to find the results for these density analysis? and section 4.3 solution overview, maybe better to move them after the feature motivations? the binning part was confusing to me until I read the appendix to see the feature design
4. figures -- the probability space analysis figure is very small, color choices might be a little too similar so it's hard to match which is which in your text to the figure, and since the points will lie on the 2-simplex, maybe adding that triangle can help us better see where the points are in the space?

**Questions:**

1. In your embedding space analysis, you claimed there's semantic overlap for TP regions and FP regions, but this is after projecting onto 2D, are we sure there is actually overlap or could there be some separating hyperplane say if we add a dimension?
2. I understand this might be out of scope, but in your embedding space analysis/probability space analysis, did any insights come up re improving recall?

---

> ### Author Response · Authors · 2025-11-24
>
> First of all thanks for reviewing our paper and providing your valuable feedback.
>
> We acknowledge that due to space constraints we added less explanations at few places and this may have caused some ambiguity in understanding the correct interpretation. We pushed many things to appendix sections, we will try to address this in the revised edition.
>
> Responses on comments:
> * With respect to equation-21 the core idea was ProbabilityDensityMap is a snapshot of the entire space as seen from a token's neighborhood perspective. The t_anchor just signifies which token's perspective is taken as the center, as the distances will change with different token centers.
> * The cross product essentially was a concise way to show that a following set of operators (mean, std, entropy, etc.) are applied on all mentioned operands (tokens, words, phrases, etc), more like a set of function name applied on function arguments, with corresponding function types.
> * We will utilize space optimally in the revised version
> * The ProbabilityDensityMap is also a feature in the Decision Tree classifier hence there are no separate results for it, there impact is reflected implicitly in NR model performance.
> * We will try to improve the figures to make them more readable
>
>
> We will now try to address the questions raised:
>
> Answer-1:
>
> There are two ways to interpret this observation. First mathematically, as described in Section 5.1, we used the Multi-Dimensional Scaling (MDS) manifold method, a global manifold-preserving algorithm to project it to a 2D space. The Metric-MDS variant minimizes the following stress function comparing pairwise distances in the original high-dimensional space ($\delta_{ij}$) with those in the low-dimensional (2D) space ($d_{ij}$):
> $$ Stress = \sqrt{\sum_{i<j}(\delta_{ij}-d_{ij})^{2}} $$
>
> The SMACOF optimization algorithm, using stress majorization (Guttman Transform), guarantees monotonic convergence. This implies if they were close in original higher dimensional space then only, they will be close in lower dimensional space. So, if there was a possibility that a hyper-plane could have separated these points then it would imply, they would be far off in original space and in that case, they would not have been mapped closed to each other in the 2D space.
>
> Secondly, from a machine learning perspective, the semantic overlap arises because NER embeddings are passed through a linear classification layer, they are not easily separable when the terms share contextual similarity. The occasional NER misclassification, despite a good F1 score, indicates issues in the input data, therefore reflects inherent semantic overlap in the embeddings.
>
> Answer-2:
>
> Improving recall is indeed important. While we considered this in our analysis, we prioritized enhancing precision in the current work and plan to address recall improvement in future research.
>
> Our findings suggest that the embedding space generated by Transformer models is inherently semantic, similar concepts are clustered together. In our analysis, regions corresponding to true biomarkers and other clinical entities were observed, though some areas were sparse or discontinuous, likely due to sample distribution. By accurately modeling the topology of this semantic space, unmarked areas could be identified and classified based on their proximity and semantic similarity to known regions. This approach could help capture additional valid entities, thereby improving overall recall in future iterations.

---

### Official Review · Reviewer_pphB · 2025-11-01

**Soundness:** 3
**Presentation:** 4
**Contribution:** 4
**Rating:** 6
**Confidence:** 2

**Summary:**

The paper builds a simple add-on to clean up mistakes from a BERT model that extracts medical terms. Instead of retraining the model, it uses a small decision tree to spot and remove uncertain predictions by looking at token probabilities and their context. This cuts false positives while keeping recall high andimproving precision for clinical text extraction.

**Strengths:**

the paper uses interpretable models (decision tree) and that makes it transparent and suitable for clinical settings. the paper also shows effectiveness across multiple datasets and hence the conlcusions can be taken as good generalization. additionally the results are strong (large reduction in false positives with minimal recall loss).

**Weaknesses:**

The quantitative analysis of htis study is great. my concern is that there is little insight into how the model behaves on more complex or ambiguous clinical text.

**Questions:**

1. Do you plan to integrate this framework with larger or newer models (e.g., ClinicalBERT or LLaMA-based models)?

2. The quantitative analysis in this study is strong, but could you provide more insight into how the model behaves on complex or ambiguous clinical text?

---

> ### Author Response · Authors · 2025-11-24
>
> First of all thanks for reviewing our paper and providing your valuable feedback.
>
> We will try to address the questions raised. We could not include all possible explanations in the main paper due to space constraints. We have added more examples in the Appendix B to showcase how these models can handle real world complex or ambiguous text. If needed we can release the annotated version of the public MIMIC-III dataset to showcase models complex clinical text handling capabilities.
>
> \textbf{Answer-1:}
> Yes, we already use these Noise Removal (NR) models alongside clinically pre-trained BERT and Llama models in a sequential pipeline. The pipeline first executes the Clinically pre-trained BERT model, followed by the NR models, and finally the Llama models.
> As described in the introduction, we pre-trained a “Custom-ClinicalBERT” using millions of clinical documents and then fine-tuned it for specific NER tasks to extract entities such as biomarkers and medications (e.g., “Custom-ClinicalBERT-NER_Biomarker,” “Custom-ClinicalBERT-NER_Medications”). These NER models were optimized for high recall with moderate precision, while the NR models were trained to remove noise and improve precision in the combined NER+NR system.
>
> Llama models provide strong reasoning capabilities but tend to hallucinate in large-scale entity extraction. Therefore, we rely on Clinically trained BERT models for controlled extraction, refined by NR models, and use Llama models for contextual reasoning e.g., determining whether a biomarker reference pertains to a specific patient finding or general information.
> Together, these three models form a high-F1 clinical extraction pipeline where BERT ensures accurate extraction, NR enhances precision, and Llama enables patient-context reasoning.
>
> \textbf{Answer-2:}
> In the Results section, we presented two tables. The first table reports performance on real-world Electronic Health Record (EHR) data from a cohort of NSCLC and breast cancer patients with active disease in 2024. These clinical notes often contain ambiguous expressions, abbreviations, misspellings, or incomplete phrases that are contextually clear to physicians but challenging for models to interpret.
>
> For example, the term EGFR may refer to Epidermal Growth Factor Receptor (a biomarker) or Estimated Glomerular Filtration Rate (a lab test). Contextual cues such as “mg/ml” or “positive” enable the models to disambiguate meaning accurately. Similarly, the term met could indicate the verb “met,” the MET biomarker, or “metastatic.”
> The results in Table 1 demonstrate the models’ ability to effectively resolve such ambiguities and achieve high F1 scores on complex, real-world clinical text.

---

### Meta-Review · Area_Chair_pzD5 · 2026-01-07

**Summary:**

This paper proposes an interpretable post processing framework for reducing false positives in clinical NER by classifying predictions as strong or weak using probability space features and a decision tree. The problem is important for clinical NLP, and reviewers agree that the approach is practical, transparent, and effective, showing large false positive reductions with minimal recall loss across multiple datasets, including cross dataset evaluation.


The main concerns relate to clarity, completeness, and positioning. Several reviewers found the presentation difficult to follow due to unclear notation, confusing feature definitions, and suboptimal organization that delays the core contribution. The paper spends excessive space on standard Transformer background while key ideas and feature motivations appear late and often require consulting the appendix. The experimental section lacks sufficient ablation studies, and analysis of behavior on more complex or ambiguous clinical text. In addition, comparisons are limited to older calibration baselines, and reviewers expect evaluation against more recent post hoc uncertainty or calibration methods.


Overall, the work is promising and practically relevant, but substantial revisions are needed to improve clarity, experimental rigor, and positioning with respect to recent literature.

**Reviewer Concerns:**

see metareview

**Reviewer Scores:**

see metareview

---

### Decision · Program_Chairs · 2026-01-26

Reject